

# Gas-phase Observations of Accretion Products from Stabilized Criegee Intermediates in Terpene Ozonolysis with Two Dicarboxylic Acids

Yuanyuan Luo[1, *], Lauri Franzon[2], Jiangyi Zhang[1], Nina Sarnela[1], Neil M. Donahue[3], Theo Kurtén[2], and Mikael Ehn[1, *]

[1] Institute for Atmospheric and Earth System Research (INAR), University of Helsinki, Helsinki, 00014, Finland
[2] Department of Chemistry, University of Helsinki, Helsinki, 00014, Finland
[3] Center for Atmospheric Particle Studies, Carnegie Mellon University, Pennsylvania, 15213, United States

*Correspondence to*: Yuanyuan Luo (yuanyuan.luo@helsinki.fi) and Mikael Ehn (mikael.ehn@helsinki.fi)

**Keywords**: Criegee intermediates, terpenes, accretion products, carboxylic acids

**Abstract.**

Criegee intermediates (CIs), forming from the ozonolysis of alkenes, are highly reactive species with diverse reaction pathways, with important roles in atmospheric chemistry. This study focuses on the formation of accretion products through reactions of thermally stabilized CIs (sCIs) from the ozonolysis of three different terpenes (α-pinene, β-pinene, and β-caryophyllene) with malonic and oxalic acids. Our experimental results demonstrate that these reactions efficiently produce the expected accretion products, though with apparent variations in yields depending on the specific terpene and acid involved. To our knowledge, these are the first direct gas-phase observations of expected adducts from terpene-derived sCIs and carboxylic acids, paving the way for a better understanding of the importance and atmospheric implications of these reactions, especially in terms of aerosol-forming capabilities of these large product molecules.



## 1. Introduction

Stabilized Criegee intermediates (sCIs), which are formed from the stabilization of excited Criegee intermediates (eCIs) during the ozonolysis of alkenes, play diverse roles in tropospheric chemistry. Depending on their structure, sCIs can have fast unimolecular reaction rates (Long et al., 2016; Chhantyal-Pun et al., 2015; Lester and Klippenstein, 2018; Drozd and Donahue, 2011), but also react with various trace atmospheric species, including water vapor, sulfur dioxide ($SO_2$), nitrogen oxides ($NO_x$), carboxylic acids, and carbonyl compounds (Cox et al., 2020; Osborn and Taatjes, 2015; Khan et al., 2018; Lin and Chao, 2017; Chhantyal-Pun et al., 2020; Gong and Chen, 2021; Taatjes et al., 2013). These reactions can produce low-volatility species that significantly contribute to the formation of both inorganic and organic aerosol components, consequently impacting air quality, climate, and human health.

Carboxylic acids are emitted directly into the atmosphere from both biogenic and anthropogenic sources and can also be produced in situ through various photochemical reactions. Observations have shown that carboxylic acids can maintain steady-state mixing ratios of up to a few parts per billion by volume (ppbv) at different locations globally (Kawamura et al., 1985; Chebbi and Carlier, 1996; Khan et al., 2015). Organic acids can react with stabilized sCIs to form adducts (Welz et al., 2014), facilitating a pathway through which alkenes are transformed into low-volatility compounds, thus contributing to the formation of secondary organic aerosol (SOA) (Chhantyal-Pun et al., 2018; Khan et al., 2018). Calculations on the reaction of formic acid with sCIs indicate that sCIs can target both the hydroxyl group and the carbonyl bond, forming stable adducts like peroxyesters, with the hydroxyl group attack being more favorable due to its higher exothermicity (Vereecken, 2017; Vereecken and Francisco, 2012). Recent direct kinetic investigations have measured reaction rate coefficients between sCIs and organic acids, finding values that greatly exceed those predicted by earlier theoretical computations or analyses of reaction end products (Chhantyal-Pun et al., 2017; Elsamra et al., 2016; Chhantyal-Pun et al., 2018). These studies have demonstrated that reactions of sCIs with carboxylic acids are typically at the gas-kinetic collision limit (Vereecken et al., 2014; Chhantyal-Pun et al., 2018).

While the significance of reactions between sCIs and organic acids has long been recognized, only recently has it become possible to measure some fundamental sCIs, enabling further experimental investigations into their properties and roles in the atmosphere (Sheps et al., 2014; Taatjes et al., 2013; Welz et al., 2014; Welz et al., 2012). Despite these advancements, the wide variety of carboxylic acids in the atmosphere, coupled with the diverse sCIs generated through the ozonolysis of alkenes, results in numerous potential reaction products. Identifying specific reaction pathways and products is especially challenging for sCIs from complex molecules like terpenes, which are a major source of atmospheric SOA (Lee et al., 2006; Yang et al., 2021; Faiola et al., 2018). Although many studies have posited that sCIs from such complex molecules reacting with organic acids could significantly contribute to the formation of SOA (Gong and Chen, 2021; Gong et al., 2024; Vansco et al., 2020; Yao et al., 2014), direct gas-phase observations of larger sCIs—such as those derived from monoterpenes and sesquiterpenes—are still extremely limited.



In our previous work on the ozonolysis of the diterpene kaurene (Luo et al., 2022), we identified a series of oxidation products containing 21-29 carbon atoms, as depicted in Figure 1(a). Initially, we hypothesized that these products might have formed from cross-reactions of peroxy radicals ($RO_2$) from kaurene with smaller $RO_2$ formed from OH oxidation of various $C_{2-9}$ organic contaminants within the chamber (Figure 1 (c)). However, we have later come to hypothesize that these might in fact have been products of sCIs with organic acid contaminants. Given that prior studies suggest a higher fraction of sCIs in larger molecules (Hakala and Donahue, 2023; Vereecken, 2017), we decided that this data is worth re-analyzing. To investigate sCI accretion products further, we conducted experiments in a flow tube reactor focusing on terpene-derived sCIs reacting with carboxylic acids. We used three different terpenes—α-pinene, β-pinene, and β-caryophyllene—and two commonly measured ambient dicarboxylic acids, malonic and oxalic acids, to explore sCIs' accretion pathways with acids across these chemical systems.

## 2. Materials and Methods

### 2.1. Laboratory Experiments

We conducted experiments in a flow reactor at room temperature under dry conditions. The total flow was set at 16 L min$^{-1}$, generated using a zero-air generator (AADCO, Series 737-14, Ohio, USA), resulting in a residence time of approximately 3 s. We utilized three different terpenes in each experiment: two monoterpenes, α-pinene and β-pinene (both $C_{10}H_{16}$), and one sesquiterpene, β-caryophyllene ($C_{15}H_{24}$). These were introduced into the flow reactor using a syringe pump setup, while ozone was added to the reactor from a Dasibi 1008-PC ozone generator. Once the terpene oxidation product signals had stabilized, we added either malonic acid ($C_3H_4O_4$) or oxalic acid ($C_2H_2O_4$) to titrate the sCIs. These dicarboxylic acids were chosen to enhance the detection of accretion products using our nitrate ion-based chemical ionization mass spectrometer (hereafter $NO_3$-CIMS). Dicarboxylic acids were chosen because they were expected to produce accretion products with functionalities that were detectable by the $NO_3$-CIMS. The acids were vaporized into the reactor by flushing $N_2$ through a vial containing the solid acid, which was heated (~60 °C) using a water bath to promote evaporation.

Ozone concentrations were monitored using a UV photometric analyzer (model 49P, Thermo Environmental). Terpene concentrations were measured by a Vocus proton-transfer-reaction time-of-flight mass spectrometer (Vocus PTR, Tofwerk AG)(Krechmer et al., 2018), which was calibrated daily with a gas-phase standard (Apel-Riemer Environmental, Inc) containing 19 components, including α-pinene. The sensitivity for α-pinene was determined as $1370 \pm 280$ cps ppb$^{-1}$. Since β-pinene and β-caryophyllene were not included in the standard, their sensitivities were estimated based on their proton affinity, fragmentation patterns, and transmission efficiencies, yielding sensitivities of $1400 \pm 289$ cps ppb$^{-1}$ for β-pinene and $1780 \pm 368$ cps ppb$^{-1}$ for β-caryophyllene, respectively. Highly oxygenated organic molecules (HOMs), accretion products, and carboxylic acids were detected using $NO_3$-CIMS (Tofwerk AG/Aerodyne Research Inc.), which has been shown to be very sensitive towards highly oxygenated species, typically with two or more H-bond donors (Junninen et al., 2010; Hyttinen et al., 2015). For such species, the charging is typically assumed to be collision limited. The $NO_3$-CIMS was not directly calibrated




in this study, owing to various challenges related to HOM quantification (Riva et al., 2019), but a calibration factor of $1\times10^{10}$ cm$^{-3}$ was used to provide a rough estimate of product concentrations. This naturally introduces a large uncertainty, estimated as at least a factor of 3, but our focus will be on species identification and on comparing relative concentration changes, meaning that the lack of detailed quantification will not hamper our conclusions. Data from all instruments were collected at a frequency of 1 Hz. Data from both the Vocus PTR-TOF and NO$_3$-CIMS were analyzed using the MATLAB tofTools package (version 612) (Junninen, 2014).

**2.2. Quantum Calculation**

The simplified mechanism of the formation of accretion product from CIs + carboxylic acid reaction is shown in Scheme 1. Quantum chemical calculations were performed to help explain the experimental observations. Reaction rates with both dicarboxylic acids were estimated for all the α-pinene and β-pinene derived sCIs, and Gibbs free energies of clustering with the nitrate ion were calculated for the accretion products derived from these. β-caryophyllene-derived sCIs were not considered due to the power law dependence of computational cost on molecular size.

In a previous computational study, L. Vereecken found that gas-phase reactions between sCI and simple carboxylic acids are collision-limited using microvariational transition state theory (Vereecken, 2017). We considered such calculations to be beyond the scope of this work due to having a high computational cost for the C$_9$ and C$_{10}$ sCIs, and simply performed a few test calculations to confirm that the sCI + dicarboxylic acid mechanism proceeds similarly to Vereecken's sCI + HCOOH mechanism, and used the dipole moment-based Structure-Activity Relationship (SAR) by Chhantyal-Pun et al. (2018) to estimate reaction rates. Detailed results and method descriptions on these calculations are found in the Supplementary.

The Gibbs free energy of clustering with the NO$_3^-$ ion for the α-pinene and β-pinene derived accretion products were calculated similarly as in Hyttinen et al. (2015). The final Gibbs free energies of clustering were compared to the corresponding value for HNO$_3$ to determine the detectability of each accretion product in our experimental conditions. The results are presented in Table S3.

All quantum chemical calculations were performed using the ORCA program, versions 5.0.4 and 6.0 (Neese, 2022, 2023).

**Scheme 1. The Criegee Intermediate + Carboxylic Acid reaction forms an α-acyloxy hydroperoxide accretion product.**



## 3. Results and Discussion

The different sCIs expected to form from the ozonolysis of the selected terpenes are summarized in Table 1. When endocyclic C=C double bonds are cleaved, the sCI will retain all the atoms from the terpene and the $O_3$, thus e.g. α-pinene produces sCIs with the formula $C_{10}H_{16}O_3$. β-pinene only features an exocyclic double bond and the cleavage will form a $C_1$ or $C_9$ sCI. According to MCMv3.3.1 (Saunders et al., 2003), the branching ratios for the formation of the $C_1$ and $C_9$ CIs are 0.4 and 0.6, respectively, while Ahrens et al. (2014) reports a branching ratio of 0.1 for $C_1$ and 0.45 each for both $C_9$ isomers based on infrared spectroscopy experiments. β-caryophyllene has both endo- and exocyclic double bonds and ozone may react with either of these bonds to form either $C_{15}H_{24}O_3$ (endocyclic case) or $CH_2O_2/C_{14}H_{22}O_2$ (exocyclic case). A previous study indicated that ozone predominantly reacts with the endocyclic bond, with the exocyclic reaction being approximately two orders of magnitude slower (Winterhalter et al., 2009).

The ozonolysis of α-pinene forms four isomeric $C_{10}$ sCI, called syn-pinonaldehyde oxide, anti-pinonaldehyde oxide, syn-isopinonaldehyde oxide, and anti-isopinonaldehyde oxide (Molecular structures shown in the supplementary). The syn- and anti-conformers are not interconvertible, but crucially for these experiments, they form the same accretion product, as the non-rotable carbonyl oxide C=O bond is converted into a rotable hydroperoxide C-O bond. Similarly, the β-pinene $C_9$ sCI has two non-convertible isomers, syn-nopinene oxide and anti-nopinene oxide, which also form the same accretion products. The latter two form the same accretion product. The β-caryophyllene derived $C_{14}$ and $C_{15}$ sCIs have not been given systematic names as far as we are aware, but the same syn-anti-isomerism applies to these as well (Winterhalter et al., 2009; Nguyen et al., 2009).

**Table 1. Main sCIs derived from the ozonolysis of the studied terpenes and their corresponding potential accretion products with malonic (MA) and oxalic (OA) acids.**

| Terpene | sCIs | Potential accretion products with MA | Potential accretion products with OA |
|---|---|---|---|
| α-pinene | $C_{10}H_{16}O_3$ | $C_{13}H_{20}O_7$ | $C_{12}H_{18}O_7$ |
| β-pinene | $CH_2O_2$ | $C_4H_6O_6$ | $C_3H_4O_6$ |
| | $C_9H_{14}O_2$ | $C_{12}H_{18}O_6$ | $C_{11}H_{16}O_6$ |
| β-caryophyllene | $CH_2O_2$ | $C_4H_6O_6$ | $C_3H_4O_6$ |
| | $C_{14}H_{22}O_2$ | $C_{17}H_{26}O_6$ | $C_{16}H_{24}O_6$ |
| | $C_{15}H_{24}O_3$ | $C_{18}H_{28}O_7$ | $C_{17}H_{24}O_7$ |

### 3.1. Accretion Products Formation in the Flow Reactor

From our previous work on kaurene ozonolysis, we identified $C_{21-29}$ products (Figure 1(a)). The primary sCI from kaurene is expected to be $C_{19}H_{30}O_2$, while we also measured $C_{2-9}$ contaminants in the chamber. If these contaminants were acids and they reacted with the $C_{19}H_{30}O_2$ sCI, several accretion products could be expected to form. To test this, we took the part of the mass



spectrum with the contaminants and shifted it by 290 Th (the mass of $C_{19}H_{30}O_2$). As depicted in Fig. 1(b), the spectrum aligns almost perfectly with the observed $C_{21-29}$ products. This supports our hypothesis that the observed species were indeed sCI + acid accretion products. In the following sections, we explore this reaction with better controlled experiments.

Figure 2 presents the mass spectra of oxidation products resulting from the ozonolysis of α-pinene, β-pinene, and β-caryophyllene, both with and without the presence of malonic acid. The concentrations of the terpenes and ozone were

maintained at similar levels before and after acid injection, ensuring comparable formation of sCIs. Prior to acid injection, we observed many of the typical HOMs associated with the different systems (Ehn et al., 2014; Jokinen et al., 2015; Richters et al., 2016a). For α-pinene, the predominant HOM monomers were $C_{10}H_{14,16}O_{7,9}$, and dimers included $C_{19}H_{28}O_9$, $C_{20}H_{30}O_{12}$, and $C_{20}H_{32}O_{9,11}$. In β-pinene ozonolysis, the major HOM monomers were $C_{10}H_{16}O_{7,8,9}$ and $C_9H_{16}O_6$, with dominant dimers such as $C_{19}H_{28}O_{9,11}$. For β-caryophyllene, the most prevalent HOM monomers were $C_{15}H_{22,24}O_{7,9}$, with dimers including $C_{29}H_{46}O_{12}$

and $C_{30}H_{46}O_{10,12,24}$.

Following the injection of malonic acid, new peaks appeared at the masses corresponding to all the expected accretion products from Table 1 (Fig. 2). Notably, no other peaks appeared, strongly indicating that sCI + acid reactions were the driver for the observed changes. For β-caryophyllene, $C_{18}H_{28}O_7$ was clearly the most abundant of the three new products, as expected, but in the β-pinene system, the signal intensity of $C_4H_6O_6$ was more than twice that of $C_{12}H_{18}O_6$. This may indicate that the

branching towards the $C_1$ CI was larger than expected, or that there are differences in the sCI + acid reaction rates or yields. It is also possible that there is difference in the sensitivity of the $NO_3$-CIMS towards the different products. For the corresponding oxalic acid experiments, we also detected all expected species at the expected masses for sCIs plus oxalic acid (Figure S1). The largest difference compared to the malonic acid system was that the signal intensities for $C_3H_4O_6$ in β-pinene and $C_{17}H_{26}O_7$ in β-caryophyllene ozonolysis were substantially higher than those of other accretion products. These differences between the

two acids were surprising, and for a more detailed look at the dependencies, we study the temporal behaviour of the different species.

Figures S2-S7 display examples of time series for the dominant $RO_2$, HOM monomers and dimers, acids, and sCIs-derived accretion products in each of the six terpene + acid systems. Comparable observations were made for the systems, and we describe below the β-caryophyllene + malonic acid system (Figure S4 (b)). Without malonic acid, upon introducing ozone into

the flow reactor with a stable β-caryophyllene concentration of approximately 30 ppb, the $O_3$-initiated $RO_2$ (e.g., $C_{15}H_{23}O_8$) appears immediately, followed by the formation of HOM monomers and dimers. The trends in $RO_2$, HOM monomers, and dimers correlate well with the ozonolysis rate of β-caryophyllene. Background malonic acid was detected in the flow reactor prior to acid injection, leading to a minor signal of $C_{18}H_{28}O_7$, the accretion product from sCI $C_{15}H_{24}O_3$ and malonic acid. Upon injecting malonic acid at around 13:55, the signal intensity of all the expected accretion products increased around tenfold,

while $RO_2$, HOM monomers, and dimers remained nearly constant, in line with earlier findings that sCI are not intermediates in HOM formation (Richters et al., 2016b). The accretion products also responded as expected to the change of β-caryophyllene and $O_3$ when the acid level was kept constant. The signal of $C_4H_6O_6$ accumulated throughout the experiment, and only decreased very slowly after the removal of β-caryophyllene and $O_3$, while all $RO_2$, HOM, as well as $C_{18}H_{28}O_7$ and $C_{17}H_{26}O_6$





fell below detection limits. A similar pattern was observed for $C_3H_4O_6$ in the oxalic acid system. We interpret this behavior as
these species being semi-volatile, thus accumulating on, and later desorbing from, the wall of the reactor or sampling lines.
Similar slow responses were observed also for malonic acid itself. In summary, the temporal trends also are in line with the
proposed sCI + acid formation mechanism.

Figure 3 illustrates the relationship between the concentration of accretion products and the ozonolysis rate of terpenes. It is
evident that all accretion products, except those from β-caryophyllene ozonolysis sCI $CH_2O_2$, track the terpene reaction rates.
The different behaviour of the two accretion products from β-caryophyllene is attributed to their semi-volatile behaviour, as
previously mentioned.

In the case of malonic acid (Figure 3(a)), α-pinene ozonolysis yielded the highest concentration of accretion product (estimated
yield of ~3%, though with large uncertainty) under similar conditions. Yields of accretion products from β-pinene-derived sCI
$CH_2O_2$ and β-caryophyllene ozonolysis-derived sCI $C_{15}H_{24}O_3$ were comparable at ~1%. Yields from β-pinene-derived sCI
$C_9H_{14}O_2$ and β-caryophyllene-derived sCI $C_{14}H_{22}O_2$ were still lower, ~0.5% and ~0.1%, respectively. However, in the oxalic
acid case (Figure 3(b)), most accretion product yields increased dramatically compared to the malonic acid case, except for
the β-pinene-derived sCI $C_9H_{14}O_2$ which decreased by a factor of 4. The highest yield was from the β-caryophyllene-derived
sCI $C_{15}H_{24}O_3$ at 50%, followed by the α-pinene-derived sCI at 30%. The total yield of adducts for β-pinene was approximately
5%.

Earlier studies have reported total sCI yields from α-pinene, β-pinene, and β-caryophyllene ozonolysis to be approximately
15%-34%(Hakala and Donahue, 2023; Sipilä et al., 2014; Zhang and Zhang, 2005), 22%-70%(Ahrens et al., 2014; Yokouchi
and Ambe, 1985; Zhang and Zhang, 2005; Hakala and Donahue, 2023), and ≥60%-74%(Nguyen et al., 2009; Winterhalter et
al., 2009; Cox et al., 2020), respectively. This pattern (i.e., β-caryophyllene > β-pinene > α-pinene) diverges from our findings
on the accretion product formation, but we also observed significant variations in the yields depending on which dicarboxylic
acid we used, with more than a factor 50 difference in the case of β-caryophyllene-derived sCI $C_{15}H_{24}O_3$. In addition, the yields
of accretion products from β-pinene-derived sCI $CH_2O_2$ + acids are higher than products from β-pinene-derived sCI $C_9H_{14}O_2$
+ acids, although previous studies have reported higher yields of $C_9$ sCI than $CH_2O_2$ sCI in β-pinene ozonolysis (Zhang and
Zhang, 2005; Ahrens et al., 2014).

There are several potential explanations for the differences between our results and earlier studies. Firstly, as illustrated in
Figure S8, there was not always sufficient acid to fully terminate all the formed sCIs to sCI-acid accretion products, though a
significant impact is only expected for the β-caryophyllene + OA case, which already showed a high yield. Another possible
explanation is differing detection sensitivities in the $NO_3$-CIMS. In our computations, the only sCI + acid accretion product
(not including the $C_{14}$-sCI products, for which calculations were not performed) with a lower detection sensitivity was one of
the two isomers of the AP-$C_{10}$ + malonic acid, which could explain why the AP-$C_{10}$ + malonic acid signal is one order of
magnitude weaker than the AP-$C_{10}$ + oxalic acid signal. However, if both accretion product isomers form in equal amounts the
signal should only decrease by a factor of two. Thus, additional explanations are needed.



The above speculations concerned experimental artifacts, but we also looked into possible structural dependencies of sCIs + acid reaction rates. Previous studies suggest that the simplest sCI reactions with carboxylic acids are barrierless and largely driven by the long-range dipole-dipole interactions (Vereecken et al., 2014; Chhantyal-Pun et al., 2018). We were able to use the dipole moment-based SAR model by Chhantyal-Pun et al. (2018) to estimate relative reaction rates for the sCI + malonic acid reactions, and conclude that these do not explain the observed differences between the AP-$C_{10}$, BP-$C_1$ and BP-$C_9$ signals, as the rate coefficients predict a formation rate ordering of BP-$C_9$ > BP-$C_1$ > AP-$C_{10}$, which is the opposite of what we see, especially when accounting for the terpene-specific total sCI yields. Furthermore, the differences in rate coefficients all fit within a factor of 1.6, which is less than the spread in product formation we see. For the sCI + oxalic acid reactions we were not able to use this model due to the latter molecule's negligible dipole moment, and we instead used collision theory to estimate the rates. While these results similarly imply minimal differences in rate coefficients, we note that the lack of dipole moment means that the relative orientation of the molecules is likely to play a larger role in determining the reaction rates. We speculate that the reduced signal of the BP-$C_9$ accretion product in the oxalic acid experiments compared to the malonic acid experiments may be due to such orientation effects, as the bicyclic BP-$C_9$ is more rigid in its internal motions than AP-$C_{10}$ or CP-$C_{14}$, and thus may be less likely to spontaneously rotate into the energy slope leading to the reaction. Lastly, differing decomposition of the products, either in the flow tube or inside the mass spectrometer, could result in fragments undetectable with these instruments.

While the ultimate reason for the differing yields estimated in our study and earlier ones remains an open question, we do stress that our main finding is that all the expected accretion products could be detected, and thus we can also look for them in the atmosphere.





**Figure 1. (a) Mass spectrum of HOMs produced by kaurene ozonolysis, with the normalized signal amplified 30-fold in the HOM dimer region (m/z > 650 Th). All peaks were detected as clusters with $NO_3^-$. (b) Mass spectrum in the m/z range that includes various species with 21-29 C-atoms. The green spectrum represents measurements after ozone injection, while the purple spectrum, indicating chamber background, is the spectrum before ozone injected (panel (c)) shifted by 290 Th (corresponding to the mass of $C_{19}H_{30}O_2$, the main sCI from kaurene ozonolysis) after the signal was reduced by a factor of 200. (c) Mass spectrum of chamber background before ozone injection.**





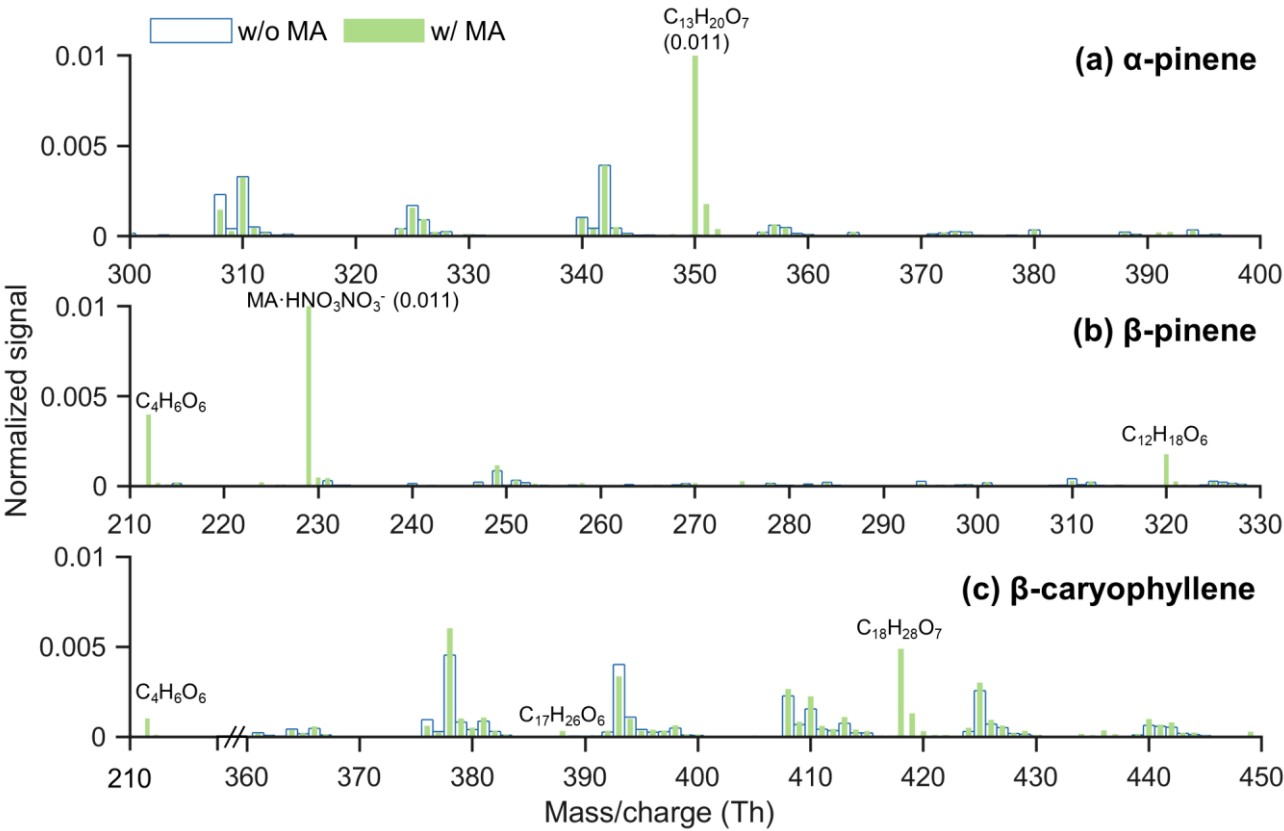

Figure 2. Mass spectrum of products formed from the ozonolysis of (a) α-pinene, (b) β-pinene, and (c) β-caryophyllene, both with and without the presence of malonic acid. New peaks are labeled with their identified compositions and numbers in parentheses denote signal strengths when peaks go off-scale.



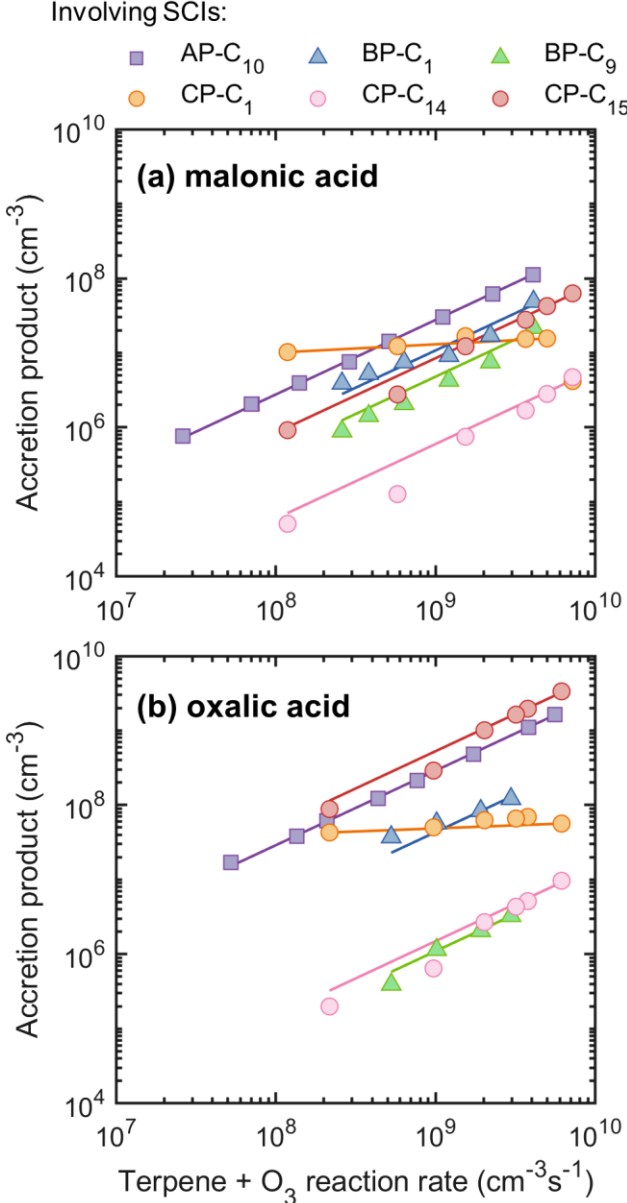

**Figure 3. Concentrations of accretion products from sCIs with (a) malonic acid and (b) oxalic acid plotted against the ozonolysis rate of terpenes. The lines represent linear fits. Legend labels indicate different accretion products formed from various sCIs in reaction with the acids. For instance, AP-C$_{10}$ in panel (a) corresponds to the accretion product C$_{13}$H$_{20}$O$_7$, formed from the sCI C$_{10}$H$_{16}$O$_3$ resulting from α-pinene ozonolysis with malonic acid; similarly, AP-C$_{10}$ in panel (b) corresponds to the accretion product C$_{12}$H$_{18}$O$_7$, formed from the same sCI reacting with oxalic acid.**





**3.2. Potential Accretion Product Observations in the Ambient Air**

A NO$_3$-CIMS is permanently deployed at the Station for Measuring Ecosystem-Atmosphere Relations (SMEAR II) in Hyytiälä, Southern Finland, to monitor low-volatility species. We analyzed data during late spring to summer (May 01 to August 31, 2023) to investigate potential accretion products from monoterpene sCIs and organic acids, as monoterpene oxidation products have been shown to dominate the HOM spectrum at this site (Yan et al., 2016; Ehn et al., 2012; Lee et al.,

2018). The time resolution of data in Hyytiälä is 30 min.

Measurements in Hyytiälä revealed a series of species whose chemical formulas corresponded to the mass of the α-pinene ozonolysis-derived sCI C$_{10}$H$_{16}$O$_3$ combined with malonic, oxalic, or formic acid. As demonstrated in Figure S9, these potential accretion products correlated well with the respective acids during the whole observation period. The most abundant species was C$_{11}$H$_{18}$O$_5$, which corresponds to the composition of the adduct of sCI C$_{10}$H$_{16}$O$_3$ and formic acid. Given that formic acid

was one of the most prevalent organic acids measured in the ambient environment (Stavrakou et al., 2012; Millet et al., 2015), its adduction with the sCI is possible, despite NO$_3$-CIMS's inability to detect formic acid directly. Malonic acid signals were significantly higher than those of oxalic acid, likely due to a higher detection efficiency of malonic acid in the NO$_3$-CIMS (Ehn et al., 2012) but the levels of C$_{13}$H$_{20}$O$_7$ and C$_{12}$H$_{18}$O$_7$ were almost comparable. Despite these observations, a good correlation (Figure S10) was also observed between these acids and various other C$_{12}$ and C$_{13}$ species. While it is possible that

the discussed species are from sCI+acid reactions, it must be kept in mind that similar diurnal cycles can be caused by many different types of processes. Consequently, we cannot definitively conclude that these species are accretion products of sCIs with organic acids.

**4. Conclusions**

This study examined the reactions between sCIs and carboxylic acids to assess their potential to form low-volatility accretion

products. Utilizing a flow reactor, we investigated the ozonolysis of three terpenes—α-pinene, β-pinene, and β-caryophyllene—combined with malonic and oxalic acids. We detected the expected accretion products from sCI + acid in all systems. Our findings indicate that accretion products from sCIs + acids formed readily, with specific variances across different terpenes and acids. For instance, α-pinene ozonolysis yielded the most significant accretion product concentration with malonic acid. Conversely, β-caryophyllene ozonolysis with oxalic acid resulted in the highest yield among the tested combinations.

These experimental results validate the hypothesis that sCIs originating from the ozonolysis of terpenes can significantly interact with organic acids forming accretion products but leave open questions regarding the yields between given sCI-acid pairs.

To our knowledge, this study is the first to show direct observations of gas-phase accretion products from terpene sCIs + acid reactions. The high efficiency of these reactions implies that sCIs could influence the dynamics and composition of organic

aerosol formation in the troposphere, but future studies should look into the rates of suppression of the accretion products at different levels of humidity.



**Data availability.** Data is available upon request by contacting the corresponding authors.

**Author Contributions.** The laboratory experiments and data analysis were conducted by YL. LF conducted the quantum calculation. JZ performed the analysis of data from ambient air. YL prepared the original draft with larger contributions from LF. All authors contributed to the discussion and comment on the manuscript.

**Competing interests.** The authors declare that they have no conflict of interest.

**Acknowledgments.** This research received support from the Academy of Finland (grant nos. 317380 and 320094), and the
China Scholarship Council (grant no. 201906220191) for YL.



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
