# Peer review of "Gas-phase Observations of Accretion Products from Stabilized Criegee Intermediates in Terpene Ozonolysis with Two Dicarboxylic Acids"

_EGUsphere, 2024_

## Author Response (AR1)

**Response to Reviewers**

**Referee 1**

The authors describe ozonolysis experiments to characterize accretion products from stabilized Criegee intermediate reactions with malonic acid and oxalic acid, and they further couple these results with field measurements that suggest evidence of the reactions occurring in the troposphere. This is a valuable contribution to understanding the range and the atmospheric impact of carbonyl oxide reactions. I have some questions and suggestions for the authors:

1.     First, there are two kinds of yields being discussed in lines 177 on, and the relationship between the two is somewhat confusing. There is the yield of the accretion product in the ozonolysis, and there is the (inferred) yield of the stabilized Criegee intermediate in the ozonolysis. These two are related in these experiments by a third yield -- the branching to the accretion product in the elementary reaction of the Criegee intermediate with the acid -- and by competing kinetic processes in the system (as described in the manuscript). I think I figured it all out, but maybe some clarifying nomenclature and a few explanatory equations would make it easier.

Response: Thank you for making us aware of our ambiguous use of yields in this section. We have tried to clarify the discussion and also explicitly added an explanation on the relationship between the two yields in Lines 190-191 and 199 – 201.

*'Although incurring large uncertainties, we can estimate the molar yields of these accretion products, i.e. the fraction of reacted terpene molecules that result in sCI + acid accretion products…*

*The yields of accretion products discussed above are determined by two key parameters: the stabilization yields of sCIs from the initial ozonolysis reaction and the rates of subsequent sCI + acid reactions. Clearly, the yield of the accretion products cannot exceed the yield of the sCIs from a given system.'*

2.     Second, I don't understand the explanation that is given for the anomalous behavior of the CH2OO reactions from the β-caryophyllene ozonolysis. The manuscript states that "these species" are semi-volatile and so the concentration doesn't respond rapidly to changes in the ozonolysis rate. But what are "these species"? If they are the accretion products of CH2OO with the acids, why don't these same compounds show that behavior when β-pinene ozonolysis is their source? Maybe I don't understand the experiment completely -- can it be explained more clearly?

Response: We appreciate the reviewer's good point. Yes, 'these products' refers to accretion products formed in reactions of $CH_2OO$ with the studied dicarboxylic acids. After thorough examination, we maintain our explanation that the anomalous behaviour of the accretion products involving $C_1$-CIs from the β-caryophyllene ozonolysis is due to SVOC characteristics. In fact, the linear fitting for BP- $C_1$-CIs accretion products, especially in the oxalic case, is not optimal compared with others. SVOCs can exhibit all kinds of unexplainable dynamics due to their complex nature. That makes us unable to fully explain the differing product behaviours of those two $C_1$-CIs accretion products from β-caryophyllene and β-pinene ozonolysis. Thus, we

opted to omit those two $C_1$-CIs accretion products from Figure 3 in main text to stick to our main points. But we still show the data in the in SI (Figure S9), as it may be useful for others who investigate the same systems in the future.

3.      Finally, from a fundamental physical chemistry point of view, it is not unexpected that these reactions would be rapid and that there would be significant branching to the stabilized product, as is supported by the computation in the supplement. The authors report CIMS measurements at Hyytiälä, which are suggestive of a detectable contribution of these reactions in the boreal forest. For other field measurements where potential signatures of similar Criegee intermediate – acid chemistry were detected (Caravan et al., Nat. Geosci. 17, 219–226 (2024)), the best estimates of the relevant emission inventories and gas phase kinetics were insufficient to explain the concentrations observed in the field. If the C11-C13 species in Figure S9 were attributable to Criegee accretion reactions, do the authors have an idea how close the kinetics of the present reactions would come to describing their field observations?

Response: Unfortunately, we are not able to make any quantitative conclusions about the importance of these sCI + organic acid reaction pathways in ambient conditions due to several limitations in available data: 1) CIs undergo multiple competing reactions in the atmosphere (e.g., with water, $SO_2$), so any concentration estimates of sCI would be extremely uncertain. 2) We do not even have quantitative data on the different monoterpene concentrations to allow a quantitative estimate of the formation rate of CI during these measurements. 3) The complex mixture of atmospheric VOCs enables multiple formation pathways that could yield products with identical elemental compositions (e.g., dimerization of $C_5$ + $C_6$ $RO_2$ forming $C_{11}$ compounds). As highlighted in Section 3.2 of our manuscript, our current data alone cannot definitively attribute these observed species solely to sCI-acid accretion products, and we would simply leave the main conclusion from the field observations to be that sCI + acid reactions may be of importance. Conversely, had we instead not detected any accretion products of this type at the site, it would have been a stronger argument for saying that they are likely not of importance.

Modeling our sCI + acid reactions in atmospheric conditions differs fundamentally from the $CH_2OO$ accretion reactions studied by Caravan et al. Their work focuses on sequential insertions of $CH_2OO$, which is produced from various sources including all organics with terminal $C=CH_2$ groups (e.g., isoprene, β-pinene, β-caryophyllene, and numerous other BVOCs). This broad range of potential $CH_2OO$ sources helps explain why current emission inventories may underestimate these reactions' importance in the Amazon. In contrast, our study examines larger sCI compounds produced specifically from select mono/sesquiterpenes, which suffers from the limitations discussed above.

4.      In addition I noticed some small typographical issues -- the Ahrens et al. reference seems to have become garbled; the rate coefficients in Table S2 are referred to as "rates" and have incorrect units.

Response: Thank you for pointing this out! The reference has been fixed, and the terminology/units have been checked and fixed in both Supplementary Information and main text.

**Referee 2**

The authors present a study of the products formed in a flow reactor during the ozonolysis of a-pinene, b-pinene, and b-caryopyllene in the presence of the dicarboxylic acids malonic acid and oxalic acid using nitrate ion chemical ionisation mass spectrometry. Re-analysis of previously reported work on the ozonolysis of kaurene is also presented in light of recent work. The paper is well presented and will be of interest to the community. I have only minor comments that should be addressed prior to publication.

1. Line 50: Are there any direct gas-phase observations of larger SCIs that can be discussed here?

**Response:** Direct gas-phase observations of larger Criegee intermediates and the adduct products from reactions between them and organic acids are extremely limited in the literature. We have discussed in Lines 50-57 of our manuscript.

*'Several papers have reported on the use of acids to scavenge sCIs (Berndt et al., 2017; Yao et al., 2014; Gong and Chen, 2021). However, these studies have primarily focused on the kinetics of sCIs and/or their role in aerosol formation, and therefore not tried to look into the formed products. Berndt et al. (2018) reported the formation of an abundant product from monoterpene sCI + acetic acid, but the data was not explicitly shown in that study. Even for smaller sCIs, very few direct observations of the expected products have been presented. Vansco et al. (2020) demonstrated the formation of functionalized hydroperoxide adducts from reactions between isoprene-derived sCIs and formic acid using multiplexed photoionization mass spectrometry.'*

2. Line 52 (and more generally): It would be helpful to include a figure, perhaps in the supplementary material, that shows the structures for the key species (such as kaurene) being discussed.

Response: Thank you for this idea. We have added a figure (Figure S1) including the structures of terpenes in the supplementary material.

3. Line 60: How do the ambient concentrations of malonic acid and oxalic acid compare to other organic acids? Which organic acids are most likely to be encountered by Criegee intermediates derived from terpenes in the atmosphere?

Response: Formic and acetic acids are the most abundant organic acids in the atmosphere (Kawamura et al., 1985; Khare et al., 1999; Paulot et al., 2011), usually within the range of 0.5 – 10 ppb (Meng et al., 1995). Malonic and oxalic acids typically exist at lower ambient concentrations compared to formic and acetic acid. They are found in the gas phase at the concentrations ranging from a few ppt to several ppb (Nah et al., 2018). As a reminder, in our experiments, we had to use dicarboxylic acids in order to make them and the products observable using our CIMS.

In general, formic and acetic acids are more likely to be encountered by CIs derived from terpenes in the atmosphere due to their higher concentrations. However, the concentrations of these organic acids in the atmosphere can vary depending on the location and environmental conditions. Thus, the most likely acids that CIs interact with can vary as well.

4. Line 93: It would be helpful to include a brief description of the calculations in the main text as well as providing detailed information in the supplementary information.

Response: A brief description was added to Lines 113-116 in the main text.

5.     Line 96: 'L. Vereecken' to 'Vereecken'.

Response: Revised.

6.     Line 101: 'found in the Supplementary' to 'found in the Supplementary Information'.

Response: Revised.

7.     Line 102: 'Gibbs free energy' to 'Gibbs free energies'.

Response: Revised.

8.     Line 120 (and elsewhere): 'syn' and 'anti' should be italicised.

Response: Revised.

9.     Line 205: It would be helpful to highlight the information provided in the supplementary information here.

Response: Included.

10.    Line 240 onwards: Are there any links that can be made with measurements reported in other work in other locations? For example, the results reported by Caravan et al. (and highlighted in the other review).

Response: Other studies have reported the detection of $C_{11}$-$C_{13}$ species in various locations worldwide (Liu et al., 2024; Frege et al., 2017; Yuan et al., 2024). However, these compounds can arise from multiple formation pathways beyond the CIs + acid reactions highlighted in our manuscript. The molecular composition alone is insufficient to conclusively determine the formation mechanism, as different precursors and chemical processes can yield compounds with identical molecular formulas. Comprehensive analytical approaches, in addition to comprehensive observations of various key parameters and compounds, are necessary before making any quantitative conclusions. Without such detailed mechanistic evidence from other field measurements, we cannot definitively link our findings to observations reported elsewhere to avoid potentially misleading.

11.    There are some formatting issues with the references

Response: Revised.

**References**

Berndt, T., Mentler, B., Scholz, W., Fischer, L., Herrmann, H., Kulmala, M., and Hansel, A.: Accretion Product Formation from Ozonolysis and OH Radical Reaction of α-Pinene: Mechanistic Insight and the Influence of Isoprene and Ethylene, Environmental Science & Technology, 52, 11069-11077, 10.1021/acs.est.8b02210, 2018.

Frege, C., Bianchi, F., Molteni, U., Tröstl, J., Junninen, H., Henne, S., Sipilä, M., Herrmann, E., Rossi, M. J., Kulmala, M., Hoyle, C. R., Baltensperger, U., and Dommen, J.: Chemical characterization of atmospheric ions at the high altitude research station Jungfraujoch (Switzerland), Atmospheric Chemistry and Physics, 17, 2613-2629, 10.5194/acp-17-2613-2017, 2017.

Kawamura, K., Ng, L. L., and Kaplan, I. R.: Determination of organic acids (C1-C10) in the atmosphere, motor exhausts, and engine oils, Environmental Science & Technology, 19, 1082-1086, 10.1021/es00141a010, 1985.

Khare, P., Kumar, N., Kumari, K., and Srivastava, S.: Atmospheric formic and acetic acids: An overview, Reviews of Geophysics, 37, 227-248, 1999.

Liu, Y., Nie, W., Qi, X., Li, Y., Xu, T., Liu, C., Ge, D., Chen, L., Niu, G., Wang, J., Yang, L., Wang, L., Zhu, C., Wang, J., Zhang, Y., Liu, T., Zha, Q., Yan, C., Ye, C., Zhang, G., Hu, R., Huang, R.-J., Chi, X., Zhu, T., and Ding, A.: The Pivotal Role of Heavy Terpenes and Anthropogenic Interactions in New Particle Formation on the Southeastern Qinghai-Tibet Plateau, Environmental Science & Technology, 58, 19748-19761, 10.1021/acs.est.4c04112, 2024.

Meng, Z., Seinfeld, J. H., and Saxena, P.: Gas/Aerosol Distribution of Formic and Acetic Acids, Aerosol Science and Technology, 23, 561-578, 10.1080/02786829508965338, 1995.

Nah, T., Ji, Y., Tanner, D. J., Guo, H., Sullivan, A. P., Ng, N. L., Weber, R. J., and Huey, L. G.: Real-time measurements of gas-phase organic acids using SF6− chemical ionization mass spectrometry, Atmospheric Chemistry and Physics, 11, 5087-5104, 10.5194/amt-11-5087-2018, 2018.

Paulot, F., Wunch, D., Crounse, J. D., Toon, G. C., Millet, D. B., DeCarlo, P. F., Vigouroux, C., Deutscher, N. M., González Abad, G., Notholt, J., Warneke, T., Hannigan, J. W., Warneke, C., de Gouw, J. A., Dunlea, E. J., De Mazière, M., Griffith, D. W. T., Bernath, P., Jimenez, J. L., and Wennberg, P. O.: Importance of secondary sources in the atmospheric budgets of formic and acetic acids, Atmospheric Chemistry and Physics, 11, 1989-2013, 10.5194/acp-11-1989-2011, 2011.

Yuan, Y., Chen, X., Cai, R., Li, X., Li, Y., Yin, R., Li, D., Yan, C., Liu, Y., He, K., Kulmala, M., and Jiang, J.: Resolving Atmospheric Oxygenated Organic Molecules in Urban Beijing Using Online Ultrahigh-Resolution Chemical Ionization Mass Spectrometry, Environmental Science & Technology, 58, 17777-17785, 10.1021/acs.est.4c04214, 2024.